# Real3D: Towards Scaling Up Large Reconstruction Models with Real-World Images

## Abstract

The default strategy for training single-view Large Reconstruction Models (LRMs) follows the fully supervised route using large-scale datasets of synthetic 3D assets or multi-view captures. Although these resources simplify the training procedure, they are hard to scale up beyond the existing datasets and they are not necessarily representative of the real distribution of object shapes. To address these limitations, in this paper, we introduce Real3D, the first LRM system that can be trained using **single-view real-world images**. Real3D introduces a novel self-training framework that can benefit from both the existing synthetic data and diverse single-view real images. We propose two unsupervised losses that allow us to supervise LRMs at the pixel- and semantic-level, even for training examples without ground-truth 3D or novel views. To further improve performance and scale the image data, we develop an automatic data curation approach to collect high-quality examples from in-the-wild images. Our experiments show that Real3D consistently outperforms prior work in four diverse evaluation settings that include real and synthetic data, as well as both in-domain and out-of-domain shapes. We will make our code, models and data available upon publication.

## 1 Introduction

The scaling law is the secret sauce of large foundation models (Kaplan et al., 2020). By scaling up both model parameters and training data, the foundation models demonstrate emerging properties and have revolutionized Natural Language Processing (Brown et al., 2020) and 2D Computer Vision (Radford et al., 2021). Recently, the same recipe has been applied to build a foundation model for *single-view 3D reconstruction*, which has the potential to benefit AR/VR (Rahaman et al., 2019), Robotics (Yan et al., 2018) and AIGC (Chen et al., 2023). From the perspective of model design, Transformers (Vaswani et al., 2017) have been proven to be effective architectures for the 2D-to-3D lifting (Jiang et al., 2023; Hong et al., 2023; Wang et al., 2023), facilitating single-view Large Reconstruction Models (LRMs) (Hong et al., 2023). From the perspective of training data, the default resources are multi-view images from 3D/video data, where we observe that increasing the dataset size (Deitke et al., 2023; 2024; Yu et al., 2023) is beneficial to the model performance. Further scaling up the LRMs is important. As single-view 3D reconstruction is an ill-posed problem due to the ambiguity from 2D to 3D (Sinha & Adelson, 1993), the essence of LRMs is learning generic shape and texture priors from large data (Hong et al., 2023).

However, the excessive reliance on multi-view supervision creates a critical bottleneck to expanding the training data of such models. The relevant strategies for data collection (i.e., synthetic 3D assets (Deitke et al., 2023) and intentional video captures (Yu et al., 2023)) are time-consuming and hard to further scale up. Thus, compared to the training data used for state-of-the-art language models (Brown et al., 2020; Touvron et al., 2023) and 2D vision models (Kirillov et al., 2023; Zhao et al., 2024), the scale of the training data for 3D reconstruction models is relatively *limited*. Moreover, the data is *biased* towards shapes that are easy to model by artists or easy to capture in the round table setting (Reizenstein et al., 2021; Yu et al., 2023), creating a domain gap between the training and inference phases.

To address this limitation, we propose to *train LRMs with single-view real-world images*. Compared to the limited number of 3D assets or intentional video captures, images are easier to collect and readily available in existing large-scale datasets (Deng et al., 2009; Thomee et al., 2016; Schuhmann

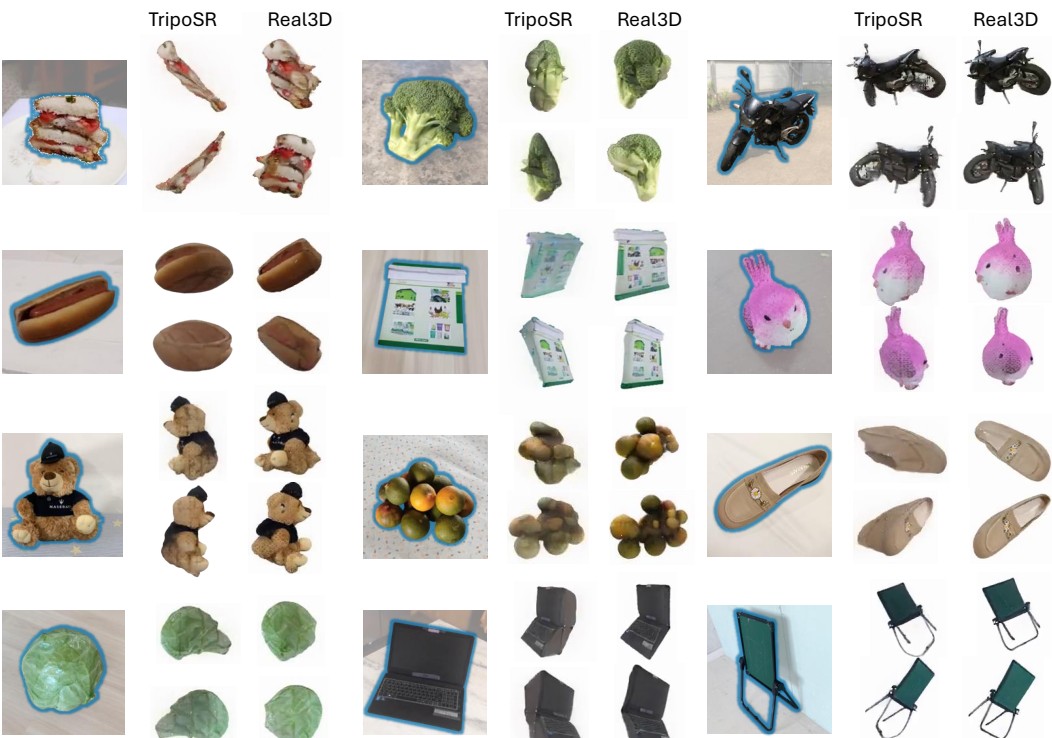

Figure 1: **Single-view 3D Reconstruction with Real3D.** We compare Real3D with the state-of-the-art TripoSR model (Tochilkin et al., 2024), which is our base model. Unlike TripoSR, which is trained solely on synthetic data, Real3D uses single-view real-world images. We provide reconstructions from two novel views, please see more results in the **supplementary video**.

et al., 2022). Furthermore, by leveraging a large number of natural images, we can capture the real distribution of object shapes more faithfully, closing the training-inference gap and improving generalization (Kuznetsova et al., 2020; Liu et al., 2021a; Yang et al., 2024). Furthermore, the maturity of recent image-based foundation models (Kirillov et al., 2023; Cho & Krähenbühl, 2023; Yang et al., 2024) makes it possible to curate these large-scale data and leverage high-quality examples, which tend to be the most beneficial for model performance.

We introduce **Real3D**, the first LRM system designed with the core principle of training on single-view real-world images. To achieve this, we propose a novel self-training framework. Our framework incorporates unsupervised training guidance, improving Real3D without multi-view real-data supervision. In detail, the guidance involves a cycle consistency rendering loss at the pixel level and a semantic loss between the input view and rendered novel views at the image level. We introduce regularization for the two losses to avoid trivial reconstruction solutions that degrade the model.

To further improve performance, we apply careful quality control for the real images that we use for training. If we naively train using all the collected image examples, the performance drops. To address this, we develop an automated data curation method for selecting high-quality examples, specifically unoccluded instances, from noisy real-world images. Eventually, we jointly perform self-training on the curated data and supervised learning on synthetic data. This strategy enriches the model with knowledge from real data while preventing divergence. Our final model consistently outperforms the state-of-the-art models that are trained without single-view in-the-wild images (Fig. 1).

We evaluate Real3D on a diverse set of datasets, spanning both real and synthetic data and covering both in-domain and out-of-domain shapes. Experimental results highlight Real3D's three key strengths: i) **Superior performance**. Real3D outperforms prior works in all evaluations. ii) **Effective use of real data**. Real3D demonstrates greater improvement using real data in single view than what the previous methods achieved using real data in multiple views. iii) **Scalability**. Performance improves as more data are incorporated, demonstrating Real3D's potential when further scaling the training data.

## 2 Related Work

**Single-view 3D Reconstruction.** Reconstructing 3D scenes and objects from a single image is a core task in 3D Computer Vision. One line of work focuses on developing better 3D representations and utilizing their unique properties to improve reconstruction quality. For example, different explicit representations have been explored, e.g., voxels (Choy et al., 2016; Tulsiani et al., 2017), point clouds (Fan et al., 2017; Jiang et al., 2018), multiplane images (Mildenhall et al., 2019; Tucker & Snavely, 2020), meshes (Liu et al., 2019; Gkioxari et al., 2019), and 3D Gaussians (Kerbl et al., 2023; Szymanowicz et al., 2023; Tang et al., 2024). Instinct representations, for example, SDFs (Park et al., 2019; Mescheder et al., 2019; Sitzmann et al., 2019), and radiance fields (Mildenhall et al., 2021; Yu et al., 2021; Rematas et al., 2021) have also improved the reconstruction accuracy.

Recent methods explore incorporating various guidance to improve the quality of single-view 3D reconstruction. MCC (Wu et al., 2023a) and ZeroShape (Huang et al., 2023) use depth guidance; however, accurate depth inputs or estimates are not always available. RealFusion (Melas-Kyriazi et al., 2023), Make-it-3D (Tang et al., 2023) and Magic123 (Qian et al., 2023) harness diffusion priors as guidance using distillation losses (Poole et al., 2022). Nevertheless, the process involves per-shape optimization, which is slow and hard to scale. To solve the problem, Zero-1-to-3 (Liu et al., 2023a) fine-tunes diffusion models for direct novel view generation. With multiple views, it is easier to get a 3D reconstruction. However, this route suffers from limited reconstruction quality, caused by inconsistency between the generated novel views (Liu et al., 2023b; Shi et al., 2023; Voleti et al., 2024). Adversarial guidance is explored to distinguish reconstruction from input view (Hu et al., 2021; Xu et al., 2022). However, adversarial training is usually unstable and difficult to extend to general categories. Moreover, semantic guidance leverages the image-to-text inversion model and the similarity calculation model to supervise novel reconstruction views (Deng et al., 2023; Tang et al., 2023). This improves the semantic consistency of the reconstructions but harms the reconstruction details, as text-to-image models lack spatial awareness (Radford et al., 2021; Gal et al., 2022). In contrast, our method, Real3D, proposes two complementary unsupervised losses to improve both the semantic and spatial consistency of our reconstruction, enabling us to train using single-view images.

**Large Reconstruction Models.** Large reconstruction models are proposed for generalizable and fast feed-forward 3D reconstruction, following the design principles of foundation models. They use scalable model architecture, e.g., Transformers (Vaswani et al., 2017; Hong et al., 2023; Jiang et al., 2023) or Convolutional U-Nets (Ronneberger et al., 2015; Tang et al., 2024; Wang et al., 2024) for encoding diverse shape and texture priors and directly mapping 2D information to 3D representations. The models are trained with multi-view rendering losses, assuming access to 3D ground-truth. For example, LRM (Hong et al., 2023) uses triplane tokens to query information from 2D image features. Other methods improve reconstruction quality by exploiting better representations (Zou et al., 2023; Wei et al., 2024; Zhang et al., 2024) and introducing generative priors (Wang et al., 2024; Xu et al., 2024; Tang et al., 2024). However, one shortcoming of these models is that they require a normalized coordinate system and canonicalized input camera pose, which limits the scalability and effectiveness of training with multi-view real-world data. To solve the problem, we enable the model to perform self-training on real-world single images, without the need for coordinate system normalization and multi-view supervision.

**Unsupervised 3D Learning from Real Images.** Learning to perform 3D reconstruction typically requires 3D ground-truth or multi-view supervision, which makes scaling up more challenging. To solve this problem, a promising avenue would be learning from massive unannotated data. Early works in this direction leverage category-level priors, where the reconstruction can benefit from category-specific templates and the definition of a canonical coordinate frame (Kar et al., 2015; Kanazawa et al., 2018; Nguyen-Phuoc et al., 2019; Wu et al., 2020; Henderson et al., 2020; Li et al., 2020; Lin et al., 2020; Monnier et al., 2022; Kulkarni et al., 2020; Duggal & Pathak, 2022; Chan et al., 2021). Recent works extend this paradigm to general categories by adjusting adversarial losses (Ye et al., 2021; Mi et al., 2022; Chan et al., 2022), multi-category distillation (Alwala et al., 2022), synergy between multiple generative models (Xiang et al., 2023), knowledge distillation (Skorokhodov et al., 2023) and depth regularization (Sargent et al., 2023). However, these methods learn 3D reconstruction from scratch, without leveraging the available 3D annotations due to limitations of their learning frameworks. Thus, their 3D accuracy and viewpoint range are limited. In contrast, our model enables initialization with available 3D ground-truth from synthetic examples and is jointly trained with in-the-wild images using unsupervised losses, which improves the reconstruction quality.

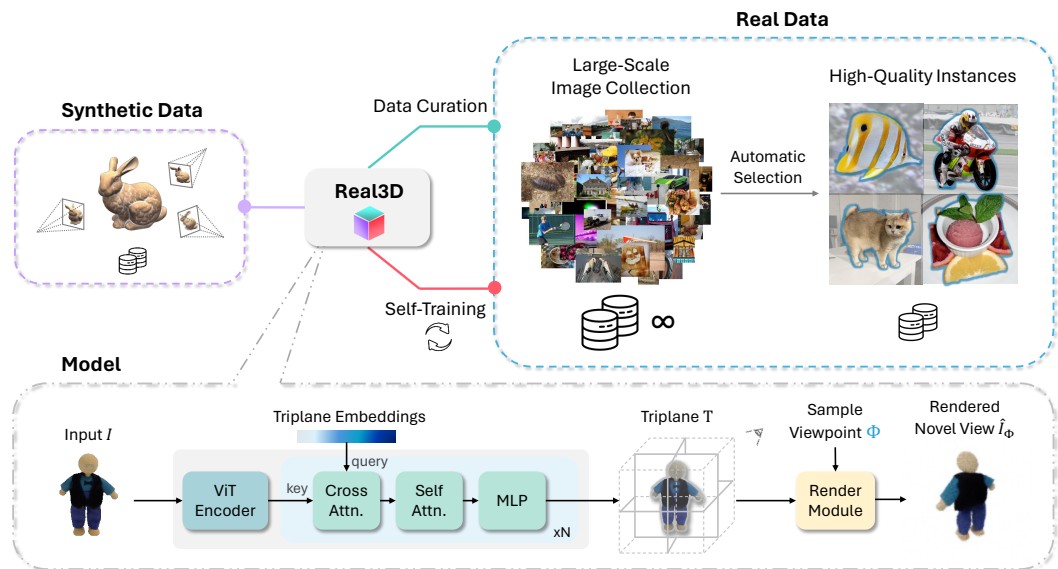

Figure 2: **Real3D overview.** (Top) Real3D is trained jointly on synthetic data (fully supervised) and on single-view real images using unsupervised losses. A curation strategy is used to identify and leverage the high-quality training instances from the initial image collection. (Bottom) We adopt the LRM model architecture.

**Model Self-Training.** Self-training helps improve performance when labeled data is limited or not available (Scudder, 1965; Radosavovic et al., 2018; Xie et al., 2020). For example, contrastive learning methods use different image augmentations to learn visual representations (He et al., 2020; Chen et al., 2020; Chen & He, 2021). This strategy has also been applied to 3D computer vision tasks, e.g., hand pose estimation (Liu et al., 2021a), detection (Yang et al., 2021), segmentation (Liu et al., 2021b) and depth estimation (Yang et al., 2024). In this paper, we propose novel losses to perform self-training on real images to improve 3D reconstruction.

## 3 PRELIMINARIES

**Large Reconstruction Model (LRM).** Let $I \in \mathbb{R}^{H \times W \times 3}$ be an input image that contains the target object to reconstruct. The LRM outputs a 3D representation of the object, $\mathbf{T} = \text{LRM}(I)$, where $\mathbf{T} \in \mathbb{R}^{3 \times h \times w \times c}$ is the latent triplane. LRM performs volume rendering (Chan et al., 2022) to produce novel views. The rendering module is formulated as $\hat{I}_\Phi = \pi(\mathbf{T}, \Phi)$, where $\hat{I}_\Phi$ is the rendered image under a target camera pose $\Phi \in \text{SE}(3)$, and $\pi$ represents the rendering process.

**Training LRM on Synthetic Multi-view Images.** Training an LRM requires multi-view image supervision. For each training object, we assume that we have several views of it with the corresponding camera poses. We denote the views and poses as $\{(I_i^{gt}, \Phi_i)|i = 1, ..., n\}$, where $n$ images are collected. This multi-view information is usually obtained from synthetic 3D assets (Deitke et al., 2023) using graphical rendering tools. The loss function on the multi-views is:

$$\mathcal{L}_{\text{RECON}}(I) = \frac{1}{n} \Sigma_{i=1}^n \left( \mathcal{L}_{\text{MSE}}(\hat{I}_{\Phi_i}, I_i^{gt}) + \lambda \cdot \mathcal{L}_{\text{LPIPS}}(\hat{I}_{\Phi_i}, I_i^{gt}) \right), \tag{1}$$

where $\lambda$ is a weight for balancing losses. The $\mathcal{L}_{\text{MSE}}$ and $\mathcal{L}_{\text{LPIPS}}$ (Zhang et al., 2018) provide pixel-level and semantic-level supervision, respectively. To facilitate training, the coordinate system needs to be normalized. More specifically, LRM assumes that i) the shape is located at the center of the world coordinate frame within a pre-defined boundary, and ii) the input view has a canonical camera pose $\phi$, a constant shared across all samples. This camera points to the world center (identity rotation) and has a constant translation vector. These assumptions can be easily satisfied with synthetic renderings.

**Training LRM on Real-World Multi-view Images.** Real-world images do not satisfy the above assumptions in most cases. To deal with the first assumption, LRM uses the sparse point cloud of COLMAP reconstruction to re-center and re-scale the world coordinate frame. Moreover, it

Figure 3: **Pixel-level Guidance using cycle-consistency**. (Left) We show the forward and backward path of the cycle. (Right) Details of the pose sampling strategy with the curriculum.

uses camera poses estimated by COLMAP to render novel views. However, this solution limits the accuracy and scalability of using real-world data, as COLMAP reconstruction can be inaccurate, and it is not trivial to capture detailed multi-view videos of objects and run COLMAP on each one of them. To deal with the second assumption, LRM is modified to condition on the input view camera pose $\phi^I$ and intrinsics $K^I$, formulated as $\mathbf{T} = \text{LRM}(I, \Phi^I, K^I)$. Note that both $\phi^I$ and $K^I$ vary across training samples, where $\phi^I$ has a *non-constant* translation vector, and $K^I$ has *non-centered* principle points due to object-region cropping. We observe that open-sourced LRM (He & Wang, 2023) can only achieve limited improvements when training with real-world multi-view data. Please see Sec. 5.3 for details.

## 4 REAL3D

We propose a novel framework that enables training LRMs using **real-world single-view images**, which are easier to collect/scale and can better capture the real distribution of object shapes. As shown in Fig. 2, we initialize an LRM on a synthetic dataset. Then we collect real-world object instances from in-the-wild images. We jointly train the model on synthetic data (Eq. 1) and perform self-training on real data. The former prevents the model from diverging with the help of supervision from ground-truth novel views. The latter introduces new data in model training, which improves the reconstruction quality and generalization capability (Sec. 4.1). We also propose an automatic data curation method to collect high-quality instance segmentations from in-the-wild images (Sec. 4.2).

### 4.1 SELF-TRAINING ON REAL IMAGES

**Overview.** The effectiveness of the multi-view training loss (Eq. 1) originates from applying supervision for improving both *pixel-level* and *semantic-level* similarity between reconstruction and ground-truth novel views. Following this philosophy, we develop novel **unsupervised** pixel-level and semantic-level guidance when training the model on single-view images, where ground-truth novel views are not available. For our base model, we use a fine-tuned TripoSR (Tochilkin et al., 2024), an LRM without input pose and intrinsics conditioning.

**Pixel-level Guidance: Cycle-Consistency.** To provide pixel-level guidance, we propose a novel cycle-consistency rendering loss. The intuition behind this is that if we have a perfect LRM, the 3D reconstruction and rendered novel views should also be perfect. Similarly, if we apply our LRM again to any of the novel views, we should be able to produce renders of the original input view perfectly. As shown in Fig. 3, we input the model with an image. We randomly sample a camera pose and render the reconstruction, obtaining a synthesized novel view. This novel view is fed back into the model to reconstruct it again. Finally, we render this second reconstruction from the viewpoint of the original input, using the inverse of the sampled pose. The goal is for this final rendering to match the original input, ensuring cycle consistency. Moreover, we observe that applying a **stop-gradient** operation on intermediate rendering can effectively prevent model degeneration and trivial reconstruction solutions. This observation is akin to self-training strategies in other domains (Chen & He, 2021).

We formulate the pixel-level guidance as follows. We input the model with an image $I^R$ that contains a shape instance in the real world and then reconstruct the latent triplane.

$$\mathbf{T}^R = \text{LRM}(I^R),\ I^R \in \mathbb{R}^{H \times W \times 3}. \tag{2}$$

We sample a camera pose $\Phi$ to render a novel view of the reconstruction and create the second input $\dot{I}_\Phi^R$ in the cycle, as:

$$\hat{I}_\Phi^R = \pi(\mathbf{T}^R, \Phi), \text{ and } \hat{I}_\Phi^R \leftarrow \text{SG}(\hat{I}_\Phi^R) \tag{3}$$

where $\text{SG}(\cdot)$ is the stop-gradient operation, and $\leftarrow$ means updating. Specifically, the sampled camera pose $\Phi$ is associated with the relative pose $\Delta\Phi$ between the sampled view and the input view, as $\Phi = \phi \cdot \Delta\Phi$, where $\phi$ is the constant canonical pose of the input view (introduced in Sec. 3).

The final image $\dot{I}^R$ that closes the cycle is formulated as:

$$\dot{I}^R = \pi(\dot{\mathbf{T}}^R, \dot{\Phi}), \text{ where } \dot{\mathbf{T}}^R = \text{LRM}(\hat{I}_\Phi^R), \text{ and } \dot{\Phi} = \phi \cdot (\Delta\Phi)^{-1}. \tag{4}$$

The pixel-level guidance is defined as $\mathcal{L}_{\text{pix}}^R = \mathcal{L}_{\text{MSE}}(\dot{I}^R, I^R)$.

However, we observe that naively applying this loss can negatively impact the model. Since the model is imperfect, any novel view it creates, especially from a camera pose significantly different from the input, can be inaccurate. This introduces errors that compound through the cycle-consistency process, ultimately degrading performance.

To address these problems, we introduce a **curriculum learning** approach. This approach progressively adjusts the complexity of the learning target from simple to difficult. Initially, the model learns from simpler cases, which later prepares it for more challenging ones. We manage the difficulty by varying the sampled camera poses from near to far relative to the original viewpoint of inputs. We formulate the camera sampling method under the curriculum as:

$$\Phi = \phi \cdot \Delta\Phi, \ \Delta\Phi \sim Uniform(-\Delta\Phi_{\text{max}}^j, \Delta\Phi_{\text{max}}^j), \tag{5}$$

where we denote *Uniform* as the uniform sampling in the SE(3) pose space, and $\Delta\Phi_{\text{max}}^j$ is the maximum sampling range of relative camera pose *at the training iteration $j$*. Note $\Delta\Phi_{\text{max}}^j$ is parameterized by the relative azimuth $\theta_{\text{max}}^j$ and elevation $\varphi_{\text{max}}^j$, which are

$$\theta_{\text{max}}^j = j/j_{\text{max}} \cdot (\theta_{\text{max}} - \theta_{\text{min}}) + \theta_{\text{min}}, \text{ and } \varphi_{\text{max}}^j = j/j_{\text{max}} \cdot (\varphi_{\text{max}} - \varphi_{\text{min}}) + \varphi_{\text{min}}, \tag{6}$$

where $j_{\text{max}}$ is the number of total training iterations. We start the curriculum with $\theta_{\text{min}} = \varphi_{\text{min}} = 15°$, and finalize with $\theta_{\text{max}} = \varphi_{\text{max}} = 90°$. The camera pose sampling range keeps increasing in the curriculum and the training process.

**Semantic-level Guidance.** The semantic guidance is performed between the novel view of the reconstruction and the input view. We leverage CLIP to compute the semantic similarity loss, as CLIP is trained on image-text pairs and can capture high-level image semantics. The image semantic similarity loss is $\mathcal{L}_{\text{CLIP}}(\hat{I}, I) = -\langle f(\hat{I}), f(I) \rangle$, where $f$ represents the visual encoder of CLIP for predicting normalized latent image features.

A simple strategy for applying the loss is rendering multiple novel views of a reconstruction and calculating the loss on all of them. However, this leads to the multi-head problem, a trivial solution to minimize the loss (see Fig. 7). The reason is that CLIP is not fully invariant to the camera viewpoint.

To handle the problem, we apply the loss to only one rendered novel view. Specifically, the novel view is least similar to the input view among multiple rendered novel views, involving hard negative mining. We also prevent rendering novel views far from the input viewpoint, as the back of the 3D shape can be semantically different from the input view.

We formulate the semantic-level guidance as follows. For the latent reconstruction $\mathbf{T}^R$ if real image data $I^R$, we render $m$ novel views using $m$ sampled rendering camera poses as:

$$\hat{I}_{\Phi_i}^R = \pi(\mathbf{T}^R, \Phi_i), \text{ for } i \in \{1, \ldots, m\} \tag{7}$$

We sample the camera poses using a similar strategy in Eq. 5 and 6, as:

$$\Phi_i = \phi \cdot \Delta\Phi_i, \ \ \Delta\Phi_i \sim Uniform(-\Delta\Phi_{\text{max}}, \Delta\Phi_{\text{max}}), \tag{8}$$

where $\Delta\Phi_{\text{max}}$ is parameterized by $\theta'_{\text{max}} = 120°$ and $\varphi'_{\text{max}} = 45°$, irrelevant to the training iteration $j$.

We calculate the semantic-level guidance as:

$$\mathcal{L}_{\text{sem}}^R = \mathcal{L}_{\text{CLIP}}(\hat{I}_{\Phi_k}^R, I^R), \text{ where } k = \underset{i \in \{1, \ldots, m\}}{\arg\max} L_{\text{CLIP}}(\hat{I}_{\Phi_i}^R, I^R). \tag{9}$$

| Input | Reconstruction | Input | Reconstruction | Input | Reconstruction |

Figure 4: **Real3D reconstruction of in-the-wild instances.** We show the input view and two novel views of the reconstruction. We use a novel front view and a novel back view. Real3D demonstrates satisfying performance on diverse shapes of in-the-wild images, especially the ones in uncommon poses.

**Training Target.** The losses applied in self-training on real data can be defined as:

$$\mathcal{L}_{\text{SELF}}^R = \lambda_{\text{in}}^R \cdot \mathcal{L}_{\text{in}}^R + \lambda_{\text{pix}}^R \cdot \mathcal{L}_{\text{pix}}^R + \lambda_{\text{sem}}^R \cdot \mathcal{L}_{\text{sem}}^R, \tag{10}$$

where $\mathcal{L}_{\text{in}}^R = \mathcal{L}_{\text{MSE}}^R(\hat{I}_\phi^R, I^R)$ is the rendering loss on the input view, and $\hat{I}_\phi^R$ is rendered using the canonical pose of input as $\hat{I}_\phi^R = \pi(\mathbf{T}^R, \phi)$. The final losses on both synthetic and real-world data can be defined as $\mathcal{L} = \mathcal{L}_{\text{RECON}}^S + \mathcal{L}_{\text{SELF}}^R$.

## 4.2 AUTOMATIC DATA CURATION

Our data curation method aims to select high-quality shape instances from real images. Specifically, we find it is important to train the model with un-occluded instances. Thus, we develop an automatic occlusion detection method leveraging the synergy between instance segmentation and single-view depth estimation. Please see Appendix B for more details.

## 5 EXPERIMENTS

We introduce our evaluation results on diverse datasets, including real images in controlled and in-the-wild settings, for comparison with previous work.

**Implementation Details.** On the synthetic data, we supervise the model on $n = 4$ renderings. On the real data, we render $m = 4$ novel views to apply the semantic guidance. We use 1 view to apply the pixel-level guidance. All evaluations are performed with a rendering resolution of 224. We set the learning rate as $4e - 5$ using the AdamW optimizer (Loshchilov & Hutter, 2017).We train the model with $j = 40,000$ iterations with a cosine learning rate scheduler. We use a batch size of 80, where we have half synthetic samples and half real-data samples. We set $\lambda$, $\lambda^R$, $\lambda_{pix}^R$, $\lambda_{sem}^R$ as 1.0, 0.3, 5.0 and 1.0. Please see more details in Appendix A.

**Datasets.** We train Real3D on our collected WildImages real single-view data and the synthetic multi-view renderings of Objaverse (Deitke et al., 2023) jointly. We evaluate the models on test splits of WildImages, MVImageNet (Yu et al., 2023), CO3D (Reizenstein et al., 2021) and OmniObject3D (Wu et al., 2023b). We introduce the details of each as follows.

• **WildImages** contains 300K single-view segmented objects, collected from datasets in diverse domains (Deng et al., 2009; Yu et al., 2023; Kuznetsova et al., 2020), which is filtered from more than 3M in-the-wild object instance segmentations. We keep aside a test split with 1000 images.

• **Objaverse** (Deitke et al., 2023) is a large-scale synthetic dataset. We use a filtered subset consisting of 260K high-quality shapes for training. We use renderings from (Qiu et al., 2023; Liu

Table 1: Evaluation results on the real-world in-domain MVImageNet dataset. We **bold** and highlight the best.

| | Eval. on Input View | | | | | | Eval. on GT Novel Views | | |
|---|---|---|---|---|---|---|---|---|---|
| | *Semantic Similarity* | | | *Self-Consistency* | | | *Novel View Synthesis Quality* | | |
| | *(Input View ↔ Rendered Novel View)* | | | *(Input View ↔ Rendered & Cycled Input View)* | | | *(GT Novel Views ↔ Rendered Novel Views)* | | |
| Method | CLIP↑ | LPIPS↓ | FID↓ | PSNR↑ | SSIM↑ | LPIPS↓ | PSNR↑ | SSIM↑ | LPIPS↓ |
| METHODS W. GENERATIVE PRIORS | | | | | | | | | |
| LGM (Tang et al., 2024) | 0.820 | 0.204 | 158.4 | 14.20 | 0.833 | 0.227 | 15.95 | 0.813 | 0.181 |
| CRM (Wang et al., 2024) | 0.823 | 0.179 | 172.5 | 15.72 | 0.873 | 0.168 | 17.54 | 0.853 | 0.142 |
| InstantMesh (Xu et al., 2024) | 0.873 | 0.188 | 153.5 | 14.81 | 0.861 | 0.171 | 14.70 | 0.806 | 0.197 |
| METHODS W/O GENERATIVE PRIORS (DETERMINISTIC) | | | | | | | | | |
| LRM (He & Wang, 2023) | 0.868 | 0.160 | 147.6 | 17.69 | 0.873 | 0.140 | 19.75 | 0.864 | 0.112 |
| TripoSR (Tochilkin et al., 2024) | 0.860 | 0.157 | 129.7 | 17.72 | 0.874 | 0.143 | 19.81 | 0.864 | 0.116 |
| **Real3D (ours)** | **0.892** | **0.147** | **116.9** | **19.80** | **0.893** | **0.125** | **20.53** | **0.871** | **0.107** |

Table 2: Evaluation results on in-the-wild images of WildImages test set. Due to the absence of ground-truth novel views, we perform evaluation on the input view.

| | Eval. on Input View | | | | | |
|---|---|---|---|---|---|---|
| | *Semantic Similarity* | | | *Self-Consistency* | | |
| Method | CLIP↑ | LPIPS↓ | FID↓ | PSNR↑ | SSIM↑ | LPIPS↓ |
| METHODS W. GENERATIVE PRIORS | | | | | | |
| LGM (Tang et al., 2024) | 0.843 | 0.188 | 146.0 | 14.13 | 0.825 | 0.210 |
| CRM (Wang et al., 2024) | 0.807 | 0.174 | 162.4 | 15.67 | 0.862 | 0.160 |
| InstantMesh (Xu et al., 2024) | 0.841 | 0.182 | 152.5 | 14.68 | 0.854 | 0.163 |
| METHODS W/O GENERATIVE PRIORS | | | | | | |
| LRM (He & Wang, 2023) | 0.847 | 0.149 | 144.9 | 18.09 | 0.872 | 0.129 |
| TripoSR (Tochilkin et al., 2024) | 0.877 | 0.148 | 128.5 | 18.18 | 0.874 | 0.125 |
| **Real3D (ours)** | **0.892** | **0.144** | **106.5** | **19.00** | **0.882** | **0.117** |

Table 3: Evaluation results on real-world out-of-domain CO3D data. We evaluate the novel view synthesis quality.

| | Eval. on GT Novel Views | | |
|---|---|---|---|
| | *Novel View Synthesis Quality* | | |
| Method | PSNR↑ | SSIM↑ | LPIPS↓ |
| METHODS W. GENERATIVE PRIORS | | | |
| LGM (Tang et al., 2024) | 15.14 | 0.802 | 0.187 |
| CRM (Wang et al., 2024) | 16.38 | 0.840 | 0.153 |
| InstantMesh (Xu et al., 2024) | 13.99 | 0.789 | 0.199 |
| METHODS W/O GENERATIVE PRIORS | | | |
| LRM (He & Wang, 2023) | 18.31 | 0.849 | 0.126 |
| TripoSR (Tochilkin et al., 2024) | 18.44 | 0.848 | 0.127 |
| **Real3D (ours)** | **19.18** | **0.855** | **0.119** |

et al., 2023a). We note this is a widely used subset (Tang et al., 2024; Qiu et al., 2023), as Objaverse contains many low-quality instances, which harm learning.

• **MVImgNet** (Yu et al., 2023), **CO3D** (Reizenstein et al., 2021) and **OmniObject3D** (Wu et al., 2023b) are used for evaluation, containing in-domain real data, out-of-domain real data, and out-of-domain synthetic data, respectively. They provide multiple views with camera pose annotations.

**Metrics.** We use two sets of metrics to evaluate the models on data with and without multi-view annotations, respectively:

• **Novel View Synthesis Metrics.** Following prior work, to evaluate reconstructions on data with multi-view information, we use standard metrics, including PSNR, SSIM (Wang et al., 2004), and LPIPS (Zhang et al., 2018).

• **Semantic and Self-Consistency Metrics.** To evaluate reconstruction quality on single-view data without ground-truth multi-view images, we introduce novel metrics. First, we render novel views of a reconstruction and measure the semantic similarity with the input view. In detail, we render 7 views where the azimuths are uniformly sampled in range $[0, 360]$ and no elevations. We use semantic metrics, including LPIPS (Zhang et al., 2018), CLIP similarity (Radford et al., 2021), and FID score (Heusel et al., 2017). Second, we evaluate the self-consistency of the reconstruction. We render a designated novel view of the reconstruction, use it as input for the LRM, and render the second reconstruction from the viewpoint of the original input. We evaluate the consistency using standard NVS metrics.

• **Mesh Quality Metrics.** We report the mesh quality using the Chamfer-L1 Distance (CD) with mesh scale 2.0. To get mesh from the triplane representation, we use the marching cube with resolution 256 for TripoSR and Real3D.

**Baselines.** We compare Real3D with LRM (Hong et al., 2023), TripoSR (Tochilkin et al., 2024), LGM (Tang et al., 2024), CRM (Wang et al., 2024) and InstantMesh (Xu et al., 2024). We use OpenLRM (He & Wang, 2023), an open-sourced LRM for comparisons. All models are trained on Objaverse (Deitke et al., 2023) unless noted otherwise. Specifically, InstantMesh uses the larger synthetic dataset Objaverse-XL (Deitke et al., 2024). Moreover, we finetune TripoSR, as it predicts reconstruction with random scales on different inputs with non-clean backgrounds, which leads to inferior evaluation results (see Appendix C) and failure of self-training. All TripoSR results are after fine-tuning. TripoSR is a base model directly comparable to ours, so we use it to ablate our contributions. For all baselines, we use the official codes and checkpoints. Furthermore, we follow the specific settings of each model to normalize target camera poses.

Table 4: Evaluation results on synthetic out-of-domain OmniObject3D data. We evaluate the novel view synthesis quality.

| | Eval. on GT Novel Views | | |
| | *Novel View Synthesis Quality* | | |
| Method | PSNR$_\uparrow$ | SSIM$_\uparrow$ | LPIPS$_\downarrow$ |
|---|---|---|---|
| METHODS W. GENERATIVE PRIORS | | | |
| LGM (Tang et al., 2024) | 15.83 | 0.791 | 0.197 |
| CRM (Wang et al., 2024) | 16.75 | 0.823 | 0.182 |
| InstantMesh (Xu et al., 2024) | 15.83 | 0.791 | 0.197 |
| METHODS W/O GENERATIVE PRIORS | | | |
| LRM (He & Wang, 2023) | 18.20 | 0.831 | 0.144 |
| TripoSR (Tochilkin et al., 2024) | 19.43 | 0.847 | 0.128 |
| **Real3D (ours)** | **20.17** | **0.855** | **0.119** |

Table 5: Ablation study on the real-world out-of-domain CO3D dataset. "Naive" means simply applying CLIP-semantic loss on all novel views . "e2e" means the cycle-consistency loss is end-to-end; "s.g." means stop the gradient of intermediate input of cycle-consistency loss.

| | $\mathcal{L}_{in}^R$ | Clean Data | Sem. Guidance | Cycle-Consistency | Curriculum | Eval. on GT Novel Views | | |
| | | | | | | *Novel View Synthesis Quality* | | |
| | | | | | | PSNR$_\uparrow$ | SSIM$_\uparrow$ | LPIPS$_\downarrow$ |
|---|---|---|---|---|---|---|---|---|
| (0) | ✗ | ✗ | ✗ | ✗ | ✗ | 18.44 | 0.848 | 0.127 |
| (1) | ✓ | ✗ | ✗ | ✗ | ✗ | 18.63 | 0.850 | 0.126 |
| | ✓ | ✓ | ✗ | ✗ | ✗ | 18.60 | 0.850 | 0.127 |
| (2) | ✓ | ✓ | ✓(naive) | ✗ | ✗ | 17.89 | 0.830 | 0.151 |
| | ✓ | ✓ | ✓($L_{sem}$) | ✗ | ✗ | 18.81 | 0.853 | 0.125 |
| (3) | ✓ | ✓ | ✓($L_{sem}$) | ✓(s.g.) | ✗ | 18.63 | 0.848 | 0.125 |
| | ✓ | ✓ | ✓($L_{sem}$) | ✓(e2e) | ✓ | 17.78 | 0.821 | 0.140 |
| | ✓ | ✗ | ✓($L_{sem}$) | ✓(s.g.) | ✓ | 18.79 | 0.852 | 0.123 |
| | ✓ | ✓ | ✓($L_{sem}$) | ✓(s.g.) | ✓ | **19.18** | **0.855** | **0.119** |

## 5.1 EXPERIMENTAL RESULTS

**Qualitative Results.** We show examples of Real3D reconstruction on in-the-wild images in Fig. 4, showing that Real3D can recover the geometry with high fidelity. Please also see Fig. 9 in Appendix D for comparisons with baselines on evaluation sets.

**Quantitative Results.** Real3D consistently outperforms prior works on all four test sets of our evaluation. As shown in Table 1 to Table 4, Real3D showcases a $0.74$ ($4\%$ relatively) PSNR improvement and a $6.3\%$ relative LPIPS improvement on average over the directly comparable TripoSR model. This demonstrates the effectiveness of our self-training method using real data. Results in Table 2 also highlight the advantage of our self-training method by using a broader data distribution.

Additionally, we observe that methods with generative priors do not perform well on out-of-distribution data. These methods generate novel views and use those views to perform sparse-view reconstruction. We conjecture that the reason is the compounding error of the novel view synthesis and reconstruction stages. This is another argument in favor of single-stage methods, like ours.

**Mesh Quality.** We report the CD for InstantMesh (the best baseline with generative prior) and TripoSR (the best deterministic baseline) are $0.395$ and $0.321$. In contrast to that, our method Real3D achieves a $0.275$ CD, improving it by $14.3\%$. We include the visualization of the mesh reconstructions in Appendix E. We observe that the other two baselines perform worse than Real3D, particularly in cases with non-common object shapes, while InstantMesh consistently struggles to faithfully reconstruct thin structures.

## 5.2 ABLATION STUDIES

**Rendering loss on input view.** As shown in Table 5 (1), when we use only the $\mathcal{L}_{in}^R$ loss, we observe slight improvements of PSNR, but SSIM and LPIPS have limited improvements. We observe a similar pattern when we add real data (raw and cleaned). This pattern implies that $\mathcal{L}_{in}^R$ can only help the model render more realistic pixels by learning the real-image pixel distribution, but it can not improve the 3D reconstruction quality.

**Semantic-level Guidance.** As shown in Table 5 (2), naively applying the CLIP-based semantic loss on all rendered novel views degrades the performance. We present visualization results in Fig. 7 of the Appendix, where we observe the multi-head problem of reconstructions. We conjecture that the reason for this observation is that copying the input view geometry to all other views is a trivial solution to minimize the semantic loss. This requires us to incorporate more regularization as we do with our semantic guidance, which achieves improvements across all metrics.

**Pixel-level Guidance.** As shown in Table 5 (3), the pixel-level cycle consistency guidance is only useful when using both clean data, stopping the gradient of intermediate rendering, and applying a training curriculum. The result demonstrates the importance of each proposed component. We present more qualitative results for the ablation in Fig. 7 of the Appendix C.

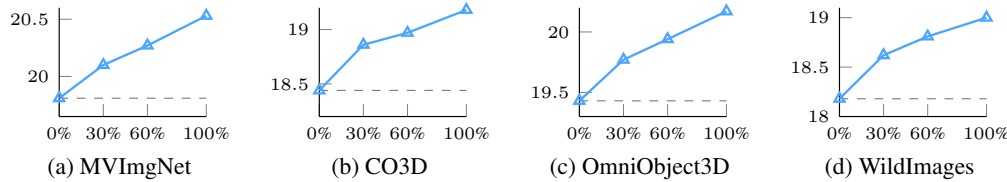

Figure 5: Real3D performance (PSNR) using different amounts of real data for training. The PSNR is evaluated on novel views for (a)-(c), and it is evaluated on (d) with self-consistency.

Table 6: Performance gain comparison between: i) using multi-view real images with multi-view rendering losses (Δmulti-view) and ii) using our single-view real images with self-training losses (Δours). Δmulti-view uses 260K MVImgNet training instances, which translates to about 8 million images. In contrast, our method uses 300K high-quality single-view images selected from about 3 million in-the-wild images for self-training.

| | Eval. on GT Novel View | | | | | | | | | Eval. on Input View | | |
| | MVImgNet | | | CO3D | | | OmniObect3D | | | WildImages | | |
| Method | PSNR↑ | SSIM↑ | LPIPS↓ | PSNR↑ | SSIM↑ | LPIPS↓ | PSNR↑ | SSIM↑ | LPIPS↓ | PSNR↑ | SSIM↑ | LPIPS↓ |
|---|---|---|---|---|---|---|---|---|---|---|---|---|
| Δmulti-view | 0.410 | 0.003 | 0.007 | 0.510 | 0.003 | 0.007 | 0.330 | 0.006 | 0.006 | 0.260 | 0.000 | 0.000 |
| Δours | **0.720** | **0.007** | **0.009** | **0.740** | **0.007** | **0.008** | **0.740** | **0.008** | **0.009** | **0.720** | **0.008** | **0.008** |

## 5.3 SCALABILITY AND EFFECTIVENESS OF SELF-TRAINING

We provide experimental results to further understand the scalability and effectiveness of our self-training method.

**Scalability - Data Amount Analysis.** Here, we evaluate the effect of scaling up the training data. As shown in Fig. 5, Real3D achieves consistent improvements as more real images are used for training. This performance gain curve demonstrates the potential for further scaling up the in-the-wild images we use for training.

**Effectiveness - Performance Gain Analysis.** We intend to further validate the effectiveness of using single-view images for self-training. For this purpose, we compare the performance gain of using self-training with another method to scale up the training data, i.e., *using captured multi-view real images*. We report the performance gain of training with multi-view images by comparing LRM variants, i.e., an LRM when trained on synthetic data only and when trained with both synthetic and real-world MVImgNet multi-view capture (He & Wang, 2023). We report this gain as Δmulti-view. We also report the gain of our self-training method as Δours. We report the results on the four datasets. As shown in Table 6, our method achieves a larger performance gain while using much fewer images for training, showing its effectiveness. The limited improvement of Δmulti-view also verifies the limitation of that training strategy at leveraging real-world multi-view data (Sec. 3). We also report the detailed numbers for computing Δmulti-view in Appendix C.

More interestingly, Δmulti-view uses only MVImgNet data for training and has limited, nearly zero, improvement on in-the-wild images. In contrast, our method achieves larger improvements by leveraging more in-the-wild data, demonstrating its efficacy in improving generalization.

## 6 CONCLUSION

We present Real3D, the first large reconstruction system that can leverage single-view real images for training. This has the major advantage of enabling training on a seemingly endless data source, that is representative of the general object shape distribution. We propose a self-training framework using unsupervised losses, which improves the performance of the model without relying on ground-truth novel views. Additionally, to further improve perfrormance, we develop an automatic data curation method to collect high-quality shape instances from in-the-wild data. Compared with previous works, Real3D demonstrates consistent improvements across diverse evaluation sets and highlights the potential of improving Large Reconstruction Models by training on large-scale image collections.

**Limitation.** One limitation of Real3D is using constant intrinsics during self-training, due to the unknown intrinsics for in-the-wild images. Although it has been proven helpful, we might observe larger improvements by incorporating an intrinsics estimation module.

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

Figure 6: The occlusion detection pipeline for data curation.

## A    MORE TRAINING DETAILS

**Training.** As we discussed, TripoSR predicts reconstruction with random scales on different inputs. The reason is that TripoSR is not conditioned on the input view camera pose and intrinsics. Thus, the model is encouraged to guess the object scale (Tochilkin et al., 2024). Moreover, TripoSR is trained on a set of Objaverse data rendered using different rendering settings. Thus, TripoSR usually overfits the training scales and can not predict the scales accurately for images rendered in different settings or in-the-wild real images. The scale of the object in the output triplane can vary and may not be consistent with regard to the scale variation in the input images. Specifically, the inaccurate scales are manifested as the misalignment between the rendered and original input view when using a canonical camera pose with a constant translation vector. And the misalignment of the scale is random.

We fine-tune TripoSR to solve the problem, using the Objaverse images rendered with a constant camera translation scale. We use a learning rate $4e - 5$ with AdamW optimizer and warmup iteration $3,000$. It is fine-tuned with $40,000$ iterations with an equivalent batch size of $80$.

For all training, we $\beta_1$, $\beta_2$, $\epsilon$ of AdamW as $0.9$, $0.96$, and $1e - 6$. We use a weight decay of $0.05$ and perform gradient clipping with the max gradient scale of $1.0$. During training, we render images with a resolution of $128 \times 128$. For each pixel, we sample $128$ points along its ray. For the images in the real dataset, we crop the instance with a random expanding ratio in $[1.45, 1.7]$ of the longer side of the instance bounding box. Our inputs have a resolution of $H = W = 512$ and the triplane has a resolution of $h = w = 64$. As TripoSR requires inputs with a gray background, we render a density mask $\hat{\sigma}_\Phi$ together with the color image $\hat{I}_\Phi^R$ when calculating the cycle-consistency pixel-level loss. We apply the rendered density mask to make the background gray. We use 8 GPUs with $48GB$ memory. It takes 4 days for training.

**Evaluation.** As our model inherits the model architecture of TripoSR, it can not handle real data input with non-centered principle points. To evaluate the models, we select a subset of MVImgNet and CO3D of 100 instances and use the image where the center of its instance mask is closest to the image center as input. We mask the background and perform center cropping with an expanding ratio of 1.6 times the mask bounding box size. We do not have any requirements for the target novel views. Besides, we use the provided COLMAP point cloud to normalize the camera poses. We normalize the input pose as the canonical pose $\phi$. The poses of other views are normalized with similarity transformations accordingly, following LRM (Hong et al., 2023). We evaluate CO3D and MVImgNet with 5 views for each instance and evaluate OmniObject3D with 10 views for each instance. To evaluate the self-consistency, we use intermediate camera pose with the azimuth of $[0, 30, 60, -30, -60, 30, 60, -30, -60, 30, 60, -30, -60]$ and elevation of $[0, 0, 0, 0, 0, 30, 30, 30, 30, 60, 60, 60, 60]$. These viewpoints cover the front of the shape, which is designed for methods with generative priors in case the generated random background influences the evaluation results for fair comparisons.

## B    DATA CURATION DETAILS

We filter the instances with three criteria. First, we filter truncated and small instances. This is achieved with simple heuristics by thresholding the instance scale and its distance to the image boundary. We use an instance scale threshold of 100 pixels and a boundary distance threshold of 10 pixels.

Table 7: Evaluation results on the real-world in-domain MVImageNet dataset. We note that LRM* is trained on both synthetic Objaverse and multi-view data of real-world MVImgeNet as an oracle comparison (results in gray). TripoSR[†] is the original TripoSR without our fine-tuning. Real3D[§] is trained on single-view images of MVImgNet without access to the multi-view information. We use 1 image for each object instance; in contrast, LRM* uses 30 multi-view images (in average) for each object. We highlight the best results. We also include the gain ($\Delta$) by using real data. $\Delta$LRM* is same as $\Delta$multi-view in Table 6.

| | Eval. on GT Novel Views | | | | | | | | |
| | *Novel View Synthesis Quality* | | | | | | | | |
| | MVImgNet | | | CO3D | | | OmniObject3D | | |
| Method | PSNR$_\uparrow$ | SSIM$_\uparrow$ | LPIPS$_\downarrow$ | PSNR$_\uparrow$ | SSIM$_\uparrow$ | LPIPS$_\downarrow$ | PSNR$_\uparrow$ | SSIM$_\uparrow$ | LPIPS$_\downarrow$ |
| LRM (He & Wang, 2023) | 19.75 | 0.864 | 0.112 | 18.31 | 0.849 | 0.126 | 18.20 | 0.831 | 0.144 |
| LRM* (He & Wang, 2023) | 20.16 | 0.867 | 0.105 | 18.82 | 0.852 | 0.119 | 18.53 | 0.837 | 0.138 |
| $\Delta$LRM* / $\Delta$multi-view | 0.410 | 0.003 | 0.007 | 0.510 | 0.003 | 0.007 | 0.330 | 0.006 | 0.006 |
| TripoSR[†] (Tochilkin et al., 2024) | 17.37 | 0.830 | 0.170 | 15.94 | 0.812 | 0.181 | 17.28 | 0.810 | 0.180 |
| TripoSR (Tochilkin et al., 2024) | 19.81 | 0.864 | 0.116 | 18.44 | 0.848 | 0.127 | 19.43 | 0.847 | 0.128 |
| **Real3D[§] (ours)** | 20.33 | 0.869 | 0.111 | 19.03 | 0.854 | 0.119 | 19.98 | 0.854 | 0.121 |
| $\Delta$ours[§] | 0.520 | 0.005 | 0.005 | 0.590 | 0.006 | 0.008 | 0.550 | 0.003 | 0.007 |

Second, We filter instances by their category. We empirically observe the LRM can not effectively reconstruct instances belonging to specific categories, e.g. bus. The reason is the large scale-variance between the front view and the side view. For example, when seeing the bus from a front view, the model can not reconstruct its side view with the correct scales, as the latent triplane representation has a cubic physical size. We observe that performing self-training on these instances harms the performance instead. We note this is a limitation of the Triplane-based LRM base model rather than our self-training framework.

Third, we filter the occluded instances. As shown in Fig. 6, we leverage the synergy between instance segmentation and single-view depth estimation for occlusion detection. We first detect the mask boundaries and then calculate the boundary parts that are contacting other instances. We use an erosion operation with kernel size 9 for boundary detection. The boundary is calculated as the difference between the eroded and the original instance mask. To detect boundaries that contact other instances, we use another erosion operation with kernel size 15. We erode the boundary of the current instance, then the contacting boundary is defined as its overlap region with the boundary of any other instances. We then determine whether an object is occluded based on whether it "owns" the boundary. For each instance, we sample $N = 20$ points (with return) on the boundary that contacts other instances. We then calculate the normal direction of the boundary at the sampled points. In detail, we use the Sobel operator to calculate the boundary tangent and normal direction. We note that we ignore the points whose 8 neighbors are all positive, during the point sampling process. We can easily know the outer and inner-mask normal directions by querying the instance mask. If the query results are both negative or positive, potentially due to the non-convex local boundary, we reject the object. Then we sample one point along each normal direction, where the sampling distance is $0.05 * s$, where $s = (b_x + b_y)/2$ and $b_x$, $b_y$ are the size of the mask bounding box in x and y-axis. We then query the estimated depth at the two sampled points, denoted as $D_{inner}$ and $D_{outer}$. If $D_{inner}/D_{outer}$ is smaller than 0.95, we consider the point occluded. If half of the sampled points on the boundary are considered occluded, they vote the object as occluded. We note that all the aggressive strategies are used to avoid false negative occlusion detection results.

We use DECOLA (Cho & Krähenbühl, 2023) and Depth Anything (Yang et al., 2024) for instance segmentation and depth estimation. We use a confidence threshold of 0.3 to filter the detection results of DECOLA. We use this low threshold for detecting all instances in the image, as any non-detected object affects the occlusion detection results. However, we empirically observe that a too-low confidence threshold, e.g. 0.1, will lead to over-segmentation and false positive detection results.

## C MORE RESULTS AND ABLATIONS

**The Model Selected to Understand Performance Gain.** In Table 6 for ablating the effectiveness of using multi-view real-world MVImgNet data, we choose OpenLRM (He & Wang, 2023) as it is the best open-source model that is trained with MVImgNet data. We note that training with multi-view

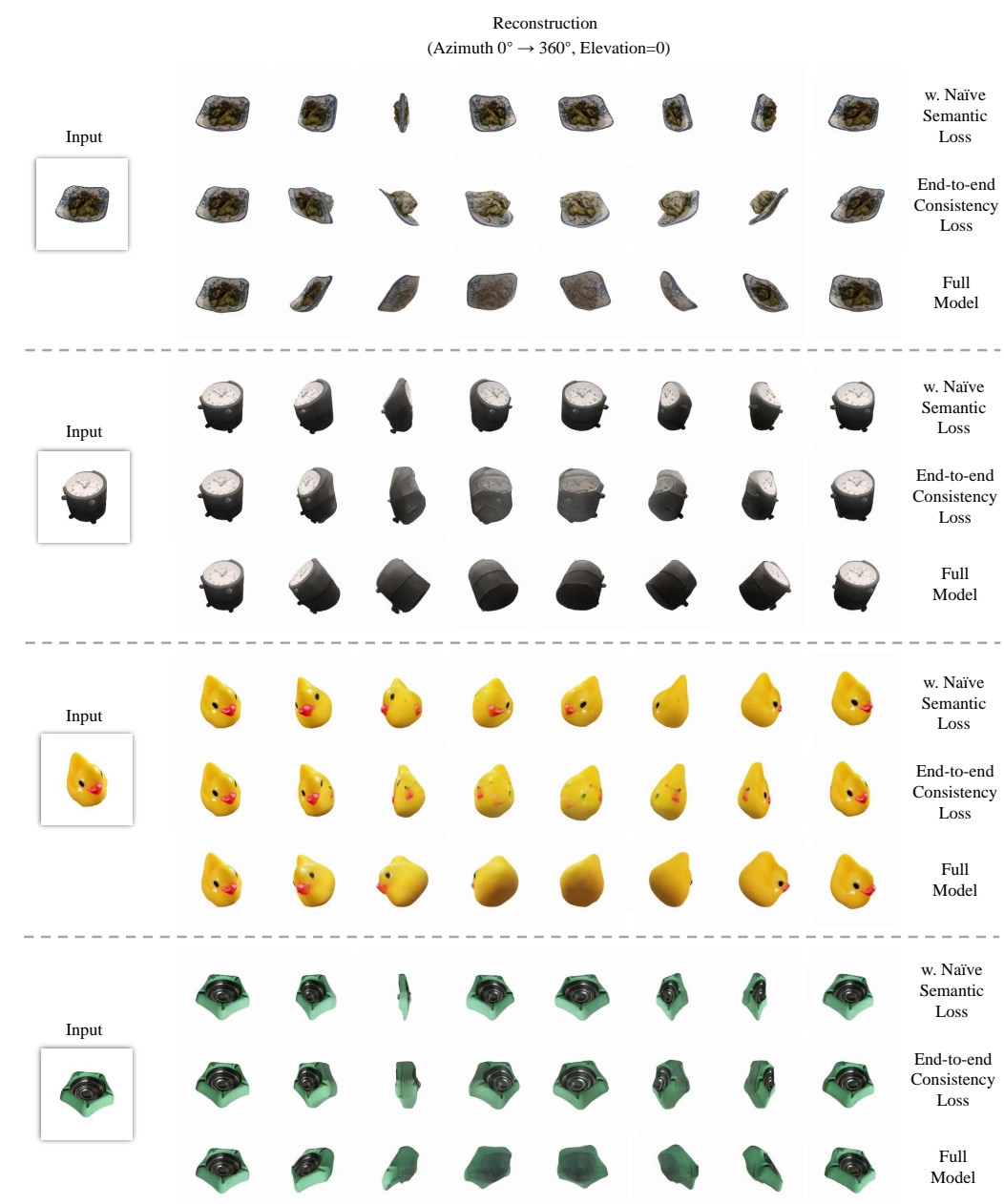

Figure 7: Visualization of ablation experiments on MVImgNet.

real-world MVImgNet data requires pose and intrinsics conditioning as we discussed in Sec. 3. Our base model TripoSR does not support this feature, as it does not have the image pose and intrinsics conditioning branch.

**Effectiveness of Self-Training.** To further evaluate the effectiveness of self-training, we compare LRM* and a Real3D version trained with MVImgNet single-view images. In this comparison, LRM* and Real3D have a similar real-world training data distribution. We note that LRM* is trained with multi-view data, where each instance of MVImgNet contains about 30 views. In contrast, Real3D only uses one image of each instance. Thus, Real3D uses the same number of shape instances for training as LRM*, but the number of training images is $30\times$ less. As shown in Table 7, Real3D outperforms LRM* and achieves larger improvements in most of the results, demonstrating the effectiveness of our self-training strategy.

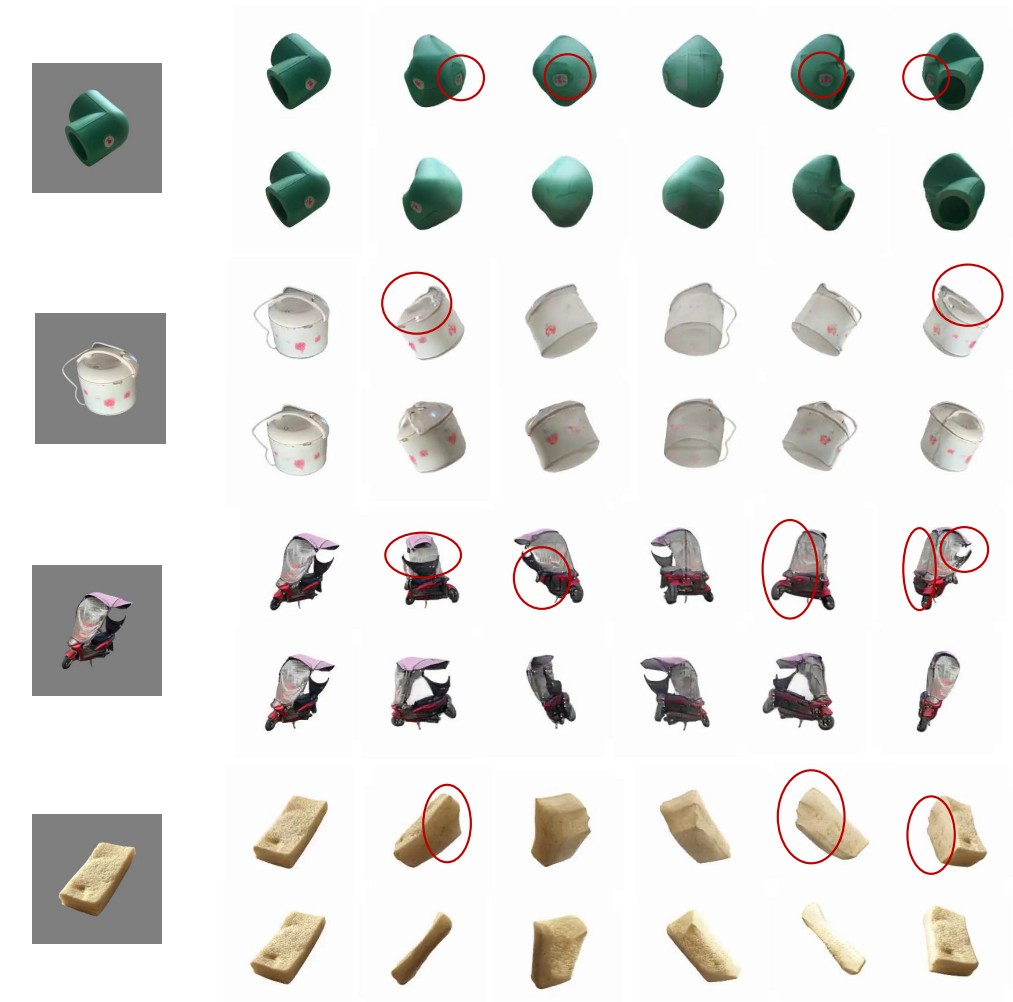

Figure 8: Qualitative comparison of the model without cycle-consistency loss (first row of each example) and with cycle-consistency loss (second row of each example).

**Original TripoSR Performance.** We also report the performance of original TripoSR (denoted as TripoSR[†]) in Table 7. Due to its random scale prediction, we observe low evaluation metrics of TripoSR[†]. We use a grid search to find the best evaluation metrics by using different camera-to-world distances.

**Visualization of Ablations.** We visualize the reconstruction of ablated models in Fig. 7 and Fig. 8. Using naive consistency loss makes the model copy the front of the object to the back of the reconstruction. Using an end-to-end cycle-consistency loss makes the reconstructions deformed in a wrong manner. Our full model can reconstruct the geometry correctly, especially the concave local geometry. Using our cycle consistency loss improves the texture and the geometry of the reconstruction, by avoiding texture leaks, unnatural and deformed shapes in general.

## D  MORE VISUALIZATION

We include additional visualization in Fig. 9. We observe that methods with generative priors usually suffer from unrealistic reconstruction, where the synthesized novel views of real objects are incorrect. This leads to the compounding error of the two-stage generation-then-reconstruction framework. Moreover, we also observe these methods usually suffer from not photo-realistic reconstruction at the back views and unaligned reconstruction content with the input images. In some other cases, they

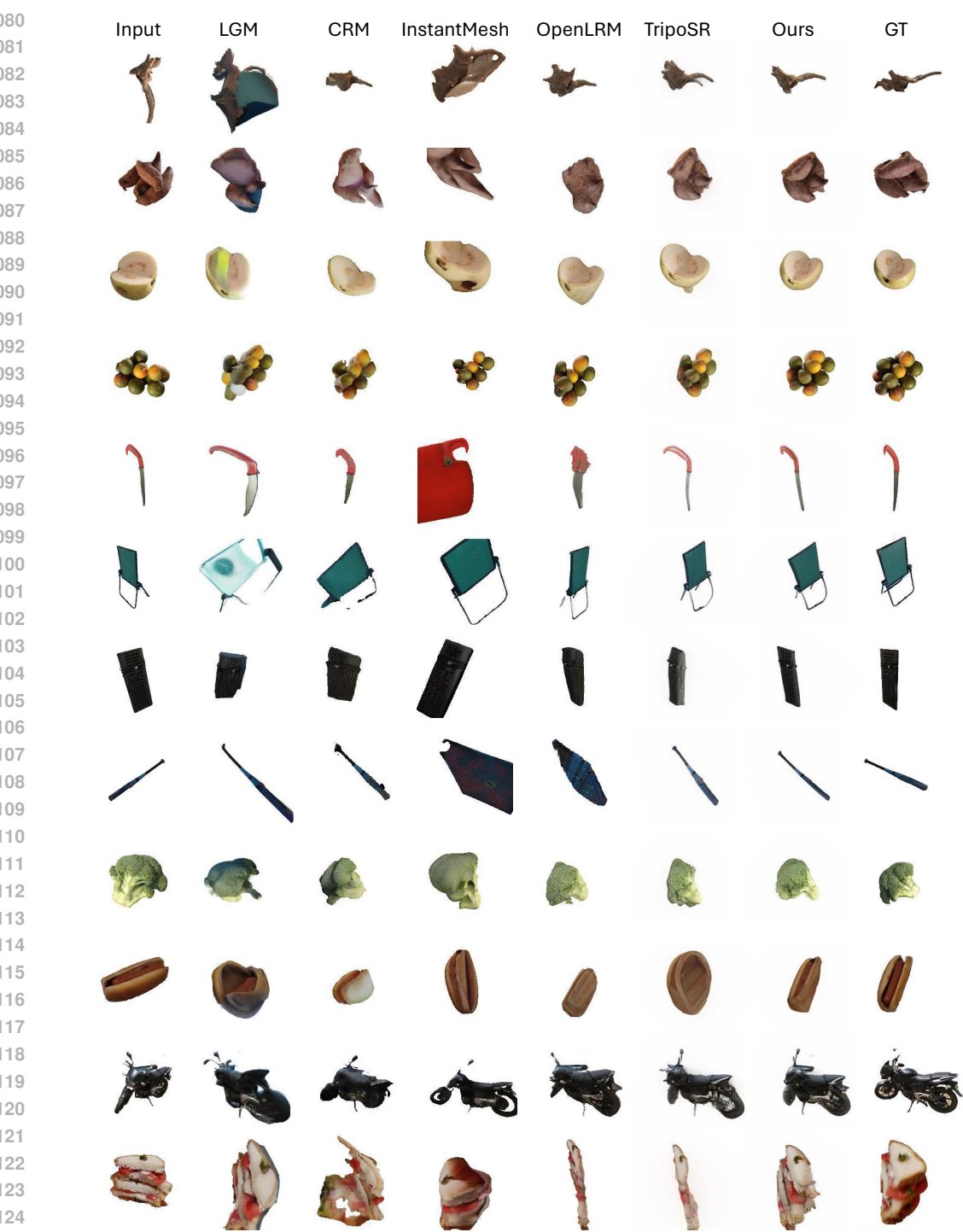

Figure 9: Visual comparison with prior works and ground-truth novel views.

can produce high-quality reconstruction, while the reconstruction content, object scale, and object pose are different from the input image.

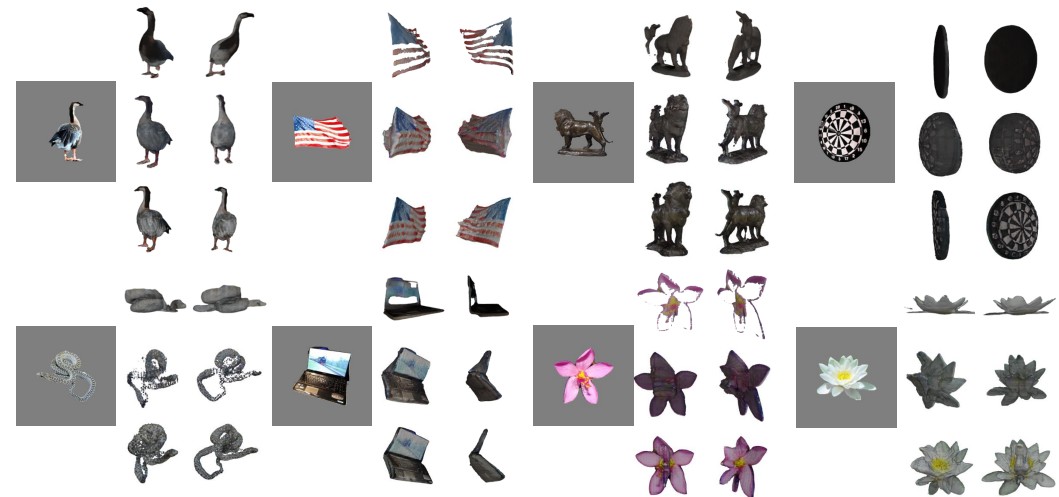

Figure 10: Visualization of meshes predicted by InstantMesh (first row of each example), TripoSR (second row of each example) and our method (last row of each example).

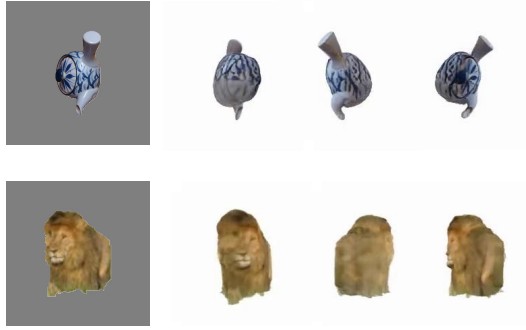

Figure 11: Representative failure cases of Real3D reconstructions.

## E   MESH VISUALIZATION

We include the mesh visualization in Fig. 10. We observe that the other two baselines perform worse than Real3D, particularly in cases with non-common object shapes, while InstantMesh consistently struggles to faithfully reconstruct thin structures.

## F   FAILURE CASES

We include the failure cases in Fig. 11. We observed that the reconstruction quality of Real3D can be compromised in cases with very unusual viewpoints, e.g., upside-down views, as well as in cases with low image quality, e.g., blurry images with potentially truncated content.

