# OpenReview forum: "Real3D: Towards Scaling Up Large Reconstruction Models with Real-World Images"
_ICLR.cc/2025/Conference — ICLR 2025 Conference Withdrawn Submission_

### Official Review · Reviewer_s4Mb · 2024-10-25

**Soundness:** 3
**Presentation:** 3
**Contribution:** 2
**Rating:** 5
**Confidence:** 3

**Summary:**

Real3D introduces a single-image training curriculum to enhance LRM, demonstrating the scalability potential of single-image datasets, which are more accessible and cost-effective to gather than 3D or video data. The experiments are sufficient, and the ablation study is comprehensive. Real3D outperforms the previous state-of-the-art, TripoSR, across multiple benchmarks, including MVImgNet, CO3D, OmniObject3D, and WildImages.

**Strengths:**

1. It's good to see that applying a self-training approach to in-the-wild single-image collections can enhance LRM's performance, improving both visual quality and generalization.

2. The experiments and ablation study are sufficient and well-rounded, further verifying the effectiveness of the proposed motivation.

3. The effort in curating high-quality data through a carefully designed detection pipeline is commendable.

4. The paper is clearly written and easy to understand.

**Weaknesses:**

- Novelty

(1) Actually, the self-supervised approach and the cycle-consistency method on single-image collections were already introduced in InfiniteNature-Zero [1], which addresses unposed scene generation, a more challenging task than this canonical-space-limited object generation setting. This related work is not even cited in the paper.

(2) The idea of using more in-the-wild data to improve generalization has been demonstrated effective in lots of previous semi-supervised approaches, such as Depth Anything [2] for depth estimation and GTA-Seg [3] for semantic segmentation. It’s not surprising that model performance improves with additional data.

(3) The ability to learn on single-image collection depends on the 3D prior of TripoSR (or other LRMs), thus this is, in essence, a two-stage training work and is more tedious to train from scratch.

- Experiment (Please also refer to "Questions" section)

(1) This comparison in this work is potentially unfair since Real3D is further fine-tuned on the baseline, TripoSR, on more collected data.

(2) The training data: Objaverse covers a wider data distribution range compared to OmniObject3D and CO3D datasets; MVImgNet and claimed collected real data may also cover the data distribution of the tested set.

[1] InfiniteNature-Zero: Learning Perpetual View Generation of Natural Scenes from Single Images. ECCV 2022 (Oral)

[2] Depth Anything: Unleashing the Power of Large-Scale Unlabeled Data. CVPR 2024

[3] Semi-Supervised Semantic Segmentation via Gentle Teaching Assistant. NeurIPS 2022

**Questions:**

(1) It would be better to provide an in-depth discussion of how Real3D, specifically for 3D object reconstruction, differs from or improves upon InfiniteNature-Zero [1] in terms of methodology.

(2) For a more fair baseline comparison, it would be ideal to fine-tune the baselines, such as TripoSR and LGM, using the same data volume, like Objaverse/CO3D/OmniObject3D multi-view renderings, and evaluate their performance on the proposed synthetic and real-world benchmarks.

(3) In Depth Anything V2 [4], the authors emphasize that data quality outweighs quantity, and they simply use synthetic data rather than real-world data in the V1 [2] version. It would be valuable to see a performance gain analysis focused on data quality, rather than solely on data quantity.

I will check other reviewer's comments and the author's feedback to adjust the paper score.


[4] Depth Anything V2.  NeurIPS 2024

---

### Official Review · Reviewer_LUWJ · 2024-11-02

**Soundness:** 2
**Presentation:** 3
**Contribution:** 2
**Rating:** 5
**Confidence:** 4

**Summary:**

1. This paper finetunes the TripoSR model using an Objaverse subset combined with a filtered monocular dataset, achieving superior results compared to TripoSR and other baselines on in- and out-domain test sets.
2. When using the monocular dataset, pixel-level and semantic-level losses are introduced to supervise novel view rendering results' pixel and semantic information. The ablation study in Table 5 demonstrates the effectiveness of finetuning with the monocular dataset and the introduction of semantic-level loss and curriculum pixel-level loss.

**Strengths:**

1. The paper clearly explains the design of different modules, and both the main text and supplementary materials provide detailed descriptions of the training and evaluation processes.
2. The experiments on "introducing monocular images as training data" are thorough, including comparisons with baseline methods on different test sets and an extensive ablation study.

**Weaknesses:**

1. The paper seems to lack discussion on the role of synthetic data loss. From my understanding, the paper finetunes the TripoSR model using an Objaverse subset and a filtered monocular dataset, but the performance gains over TripoSR may stem from selected Objaverse subset and the monocular dataset.
Specifically, Table 5 analyzes the improvements brought by the monocular dataset (using Equ. 10). For instance, introducing clean data loss improves PSNR from 18.44 to 18.60, adding semantic guidance further increases PSNR to 18.81, and introducing curriculum pixel-level constraints raises it to 19.18. However, the analysis only focuses on $L_{SELF}$ as defined in Equ. 10. The question remains: if we finetune TripoSR using only the Objaverse subset (with $L_{RECON}$ only), will there still be improvements compared to TripoSR? If so, how do these gains compare to those from $L_{SELF}$?
Another example is the comparison between Real3D and TripoSR in Table 2, which shows limited improvements over TripoSR. How much of these gains are due to $L_{RECON}$, and how much is due to $L_{SELF}$ ?

**Questions:**

Please refer to the Weaknesses section. If the authors could provide further clarification on these points, I would be open to reconsidering my rating. Thank you.

---

### Official Review · Reviewer_QScn · 2024-11-03

**Soundness:** 3
**Presentation:** 3
**Contribution:** 2
**Rating:** 5
**Confidence:** 4

**Summary:**

This paper addresses the limitations of current training strategies for large reconstruction models (LRMs), which typically rely on large-scale, multi-view datasets. The authors propose a method to expand the training data to include single-view images from real-world settings, which are easier to acquire. To achieve this, they introduce a self-training framework that can leverage both existing synthetic data and a diverse set of single-view images. For these single-view images, the authors incorporate a cycle consistency constraint and a curriculum sampling strategy during training. Additionally, they enhance supervision by applying CLIP similarity across multi-view images. To ensure suitable training data, they present an automatic data curation method that selects high-quality examples from in-the-wild images. Extensive experiments across different datasets are carried out to validate the approach.

**Strengths:**

1. The paper is well-motivated, focusing on a key limitation of current LRM methods, which rely heavily on large-scale 3D datasets like Objaverse. This reliance restricts their applicability and caps their performance potential. Learning 3D structures from unposed images in the wild through unsupervised methods has been a long-standing challengel.

2. In light of the lack of multi-view ground truth supervision, the proposed method reasonably incorporates cycle consistency and curriculum sampling on camera pose during training, alongside an automated data curation approach.

3. The paper presents extensive experimental results across a diverse range of datasets, covering both synthetic and real-world data.

**Weaknesses:**

1. The evaluation metrics are suboptimal. The paper employs PSNR, LPIPS, and SSIM, which are mainly designed for image reconstruction tasks. For single-image-to-3D generation, the unseen parts of an object inherently involve uncertainty and diversity, meaning there should not be a single, fixed ground truth for 3D shape or novel view images. Although CLIP scores are provided in some tables, this metric is also vague, and cannot reveal the image quality. The paper would benefit from including more generative-based metrics, such as FID, or conducting a human study or GPT-based evaluation.

2. The performance improvements shown by the current metrics are minimal, which may also relate to the issue noted above. An average PSNR below 20 suggests that the generated and target images are very dissimilar, so the small gap between methods may not reveal meaningful differences.

3. The ablation study in Table 5 is not fully convincing; it does not clearly demonstrate that each component has a consistently positive impact on performance. For example, in some cases, the inclusion of multiple modules results in lower scores than the baseline.

4. The semantic-level supervision relies on multi-view image CLIP embedding similarity; however, CLIP embeddings do not guarantee multi-view invariance. Although the authors state they avoid rendering novel views far from the input viewpoint, this approach represents a trade-off: either sampling views close to the input, which minimizes the intended effect, or introducing erroneous rotational symmetry that artificially increases multi-view CLIP similarity. Additionally, the ablation studies in Table 5 do not clearly demonstrate the benefits of this loss (full model vs. full model without segmentation guidance).

5. The data curation process lacks qualitative or quantitative evaluation. More examples or results would be helpful to illustrate the advantages of this process.

**Questions:**

Please refer to the weakness section.

---

### Official Review · Reviewer_c7co · 2024-11-03

**Soundness:** 3
**Presentation:** 3
**Contribution:** 2
**Rating:** 5
**Confidence:** 4

**Summary:**

This paper present a novel method to train LRM on real world dataset. Because most of real world image dataset do not contain multiview images, authors introduce a novel method using cycle-consiste This paper proposes an innovative approach for training Large Reconstruction Models (LRMs) on real-world datasets. Since most real-world image datasets lack multi-view images, the authors present a novel technique leveraging cycle-consistency to overcome this limitation.

**Strengths:**

1. The paper is well-structured, with a logical flow and a strong, clearly articulated motivation.

2. The experiments are comprehensive, demonstrating the model's potential for scalability.

3. The use of curriculum learning facilitates easier training, and the model has been tested extensively on diverse real-world datasets such as MVImgNet, CO3D, and OmniObject3D.

**Weaknesses:**

1. The LRM model has already been trained on MVImgNet. This raises questions for me about its novelty.

2. The performance improvements appear marginal, as demonstrated by the examples shown in Figure 1. Additionally, Tables 3 and 4 shows that Real3D performs similarly to TripoSR. I have yet to see impressive performance gains from incorporating real-world data into the training process, and this casts doubt on the Real3D scalability potential.

3. The results of Real3D seem sensitive to the elevation angles of input images, an issue that also affects the base LRM. Real-world datasets often contain images with significant elevation variations, unlike synthetic datasets where images are usually rendered with lower elevation angles for training. It would be valuable to explore strategies to address this challenge in future research.

**Questions:**

Same as in the Weakness.

---

### Note · Authors · 2024-11-14

I have read and agree with the venue's withdrawal policy on behalf of myself and my co-authors.